# Viral delivery of tissue nonspecific alkaline phosphatase diminishes craniosynostosis in one of two FGFR2$^{C342Y/+}$ mouse models of Crouzon syndrome

Hwa Kyung Nam, Iva Vesela, Sara Dean Schutte, Nan E. Hatch *

Department of Orthodontics and Pediatric Dentistry, School of Dentistry, University of Michigan, Ann Arbor, Michigan, United States of America

* nhatch@umich.edu

**Data Availability Statement:** The data underlying the results presented in the study are available from the University of Michigan's Deep Blue

## Abstract

Craniosynostosis is the premature fusion of cranial bones. The goal of this study was to determine if delivery of recombinant tissue nonspecific alkaline phosphatase (TNAP) could prevent or diminish the severity of craniosynostosis in a C57BL/6 FGFR2$^{C342Y/+}$ model of neonatal onset craniosynostosis or a BALB/c FGFR2$^{C342Y/+}$ model of postnatal onset craniosynostosis. Mice were injected with a lentivirus encoding a mineral targeted form of TNAP immediately after birth. Cranial bone fusion as well as cranial bone volume, mineral content and density were assessed by micro CT. Craniofacial shape was measured with calipers. Alkaline phosphatase, alanine amino transferase (ALT) and aspartate amino transferase (AST) activity levels were measured in serum. Neonatal delivery of TNAP diminished craniosynostosis severity from 94% suture obliteration in vehicle treated mice to 67% suture obliteration in treated mice, p<0.02) and the incidence of malocclusion from 82.4% to 34.7% (p<0.03), with no effect on cranial bone in C57BL/6 FGFR2$^{C342Y/+}$ mice. In contrast, treatment with TNAP increased cranial bone volume (p< 0.01), density (p< 0.01) and mineral content (p< 0.01) as compared to vehicle treated controls, but had no effect on craniosynostosis or malocclusion in BALB/c FGFR2$^{C342Y/+}$ mice. These results indicate that postnatal recombinant TNAP enzyme therapy diminishes craniosynostosis severity in the C57BL/6 FGFR2$^{C342Y/+}$ neonatal onset mouse model of Crouzon syndrome, and that effects of exogenous TNAP are genetic background dependent.

## Introduction

Skull growth occurs via intramembranous bone deposition along the outer edge of each cranial vault bone, in coordination with endochondral growth of cranial base bones at the cranial base synchondroses (cartilaginous growth plates). Anterior-posterior growth of the skull is dependent upon both cranial vault and cranial base bone growth. Craniosynostosis (which, when syndromic also often includes premature fusion of cranial base bones) can lead to high

Database; https://deepblue.lib.umich.edu/data/collections/7m01bk78r?locale=en.

**Funding:** This work was supported by grant R01DE02582701. to N.E.H. from the National Institute of Dental and Craniofacial Research (NIDCR). The funders played no role in the study design, data collection, data analysis, decision to publish, or preparation of the manuscript.

**Competing interests:** The authors have declared that no competing interests exist.

intracranial pressure, abnormal skull and facial shapes, malocclusion, blindness, seizures and brain abnormalities [1–6]. Because the sole treatment is surgery, even with appropriately early diagnosis patients can suffer high morbidity [7–9]. A pharmaceutical treatment for craniosynostosis is not yet realized.

Activating mutations in Fibroblast Growth Factor Receptor 2 (*Fgfr2*) [10–13] and inactivating mutations in *Alpl*, the gene for tissue nonspecific alkaline phosphatase (TNAP) [14–17] can cause craniosynostosis. In this study we sought to determine if treatment with recombinant mineral-targeted TNAP could rescue craniosynostosis and associated craniofacial skeletal abnormalities in the FGFR2$^{C342Y/+}$ Crouzon mouse model of craniosynostosis, when delivered shortly post-natal with lentivirus. The FGFR2$^{C342Y/+}$ mutation was previously demonstrated to cause ligand independent signaling and is therefore widely considered to be an activating mutation leading to increased FGF signaling [18–21]. We pursued this investigation because we previously demonstrated that FGF signaling diminishes TNAP expression [22–24], and showed that TNAP deficiency in mice leads to a similar craniofacial phenotype to that seen in FGFR2$^{C342Y/+}$ Crouzon mice including coronal but not sagittal craniosynostosis, deficient growth of the cranial base with fusion of cranial base synchondroses, and brachycephalic/acrocephalic (wide/tall) head shapes [11, 17, 25, 26]. Additionally, in a previous study using archival aliquots of lentivirus expressing the mineral targeted recombinant form of TNAP that resulted in increases in serum AP activity in only a small number of the treated mice, we found significant differences in the morphology of the inferior skull surface in treated vs. untreated BALB/c FGFR2$^{C342Y/+}$ mice [27].

Craniosynostosis onset in humans can occur pre- or postnatal, with earlier onset forms leading to more severe outcomes and higher morbidity. We backcrossed FGFR2$^{C342Y/+}$ mice onto C57BL/6 and BALB/c strains, and found that C57BL/6 FGFR2$^{C342Y/+}$ mice exhibit obliteration of the coronal cranial suture initiating shortly after birth, while BALB/c FGFR2$^{C342Y/+}$ mice exhibit point fusions across the coronal suture initiating at four weeks after birth [25]. Both FGFR2$^{C342Y/+}$ strains of mice also exhibit deficient cranial base growth with brachycephalic/acrocephalic head shapes and tendency for a class III malocclusion (lower teeth protruding anterior to upper teeth). The objective of this study was to determine if postnatal delivery of recombinant TNAP could diminish the severity of craniosynostosis in BALB/c FGFR2$^{C342Y/+}$ mice, a model of less severe Crouzon craniosynostosis syndrome and/or in C67BL/6 FGFR2$^{C342Y/+}$ mice, a model of more severe Crouzon craniosynostosis syndrome. Viral delivery of TNAP was tested in both C57BL/6 FGFR2$^{C342Y/+}$ and BALB/c FGFR2$^{C342Y/+}$ mice to determine if efficacy is dependent upon timing of craniosynostosis onset and/or severity of fusion.

## Materials and methods

### TNAP lentivirus

Recombinant mineral-targeted TNAP lentivirus was generously provided by Dr. Jose Luis Millán (Sanford Burnham Prebys Medical Discovery Institute, La Jolla, CA). Pseudotype of the virus is VSV-G, for transduction into multiple cell types. The SJ1-based HIV-1 vector contains 0.25 kB insulators derived from the chicken beta-globin locus to diminish insertional mutagenesis, increase viral titer and increase protein expression [28]. The virus expresses a mineral-targeted protein that is composed of soluble human TNAP enzyme fused to the constant region of human IgG1 and a C-terminal deca-aspartate motif to confer targeting to hydroxyapatite. The aspartate tag confers 30x higher affinity for hydroxyapatite than untagged enzyme [29]. Longevity of the virus for increasing AP levels was reported to last at least 60 days when injected into neonatal mice and biodistribution of the virus was previously reported to be high

in the liver [30]. Treatment with this recombinant form of TNAP was previously shown to increase serum alkaline phosphatase levels and rescue bone defects seen in hypophosphatasia [14, 30–33]. Production and titer of the lentivirus for this study was performed by the University of Michigan Vector Core.

### Animal procedures

Because severity of craniosynostosis and associated craniofacial shape defects are variable on the mixed genetic background, FGFR2$^{C342Y/+}$ mice were backcrossed with BALB/c and C57BL/ 6 mice (obtained from Charles River Laboratories) for at least fifteen generations prior to experiments. BALB/c FGFR2$^{C342Y/+}$ mice have a more moderate form of Crouzon syndrome with craniosynostosis in the form of point fusions across the coronal suture first apparent between three and four weeks after birth [25]. C57BL/6 mice have a more severe form of Crouzon syndrome with craniosynostosis in the form of suture obliteration first apparent in neonatal mice (Fig 1). Both mice also exhibit abnormalities in the cranial base and deficient cranial base growth. Genotyping was performed as previously described [11, 25]. Briefly, DNA from tail digests was amplified by polymerase chain reaction using `5'-gagtaccatgctgactg catgc-3'` and `5'-ggagaggcatctctgtttcaagacc-3'` primers to yield a 200 base

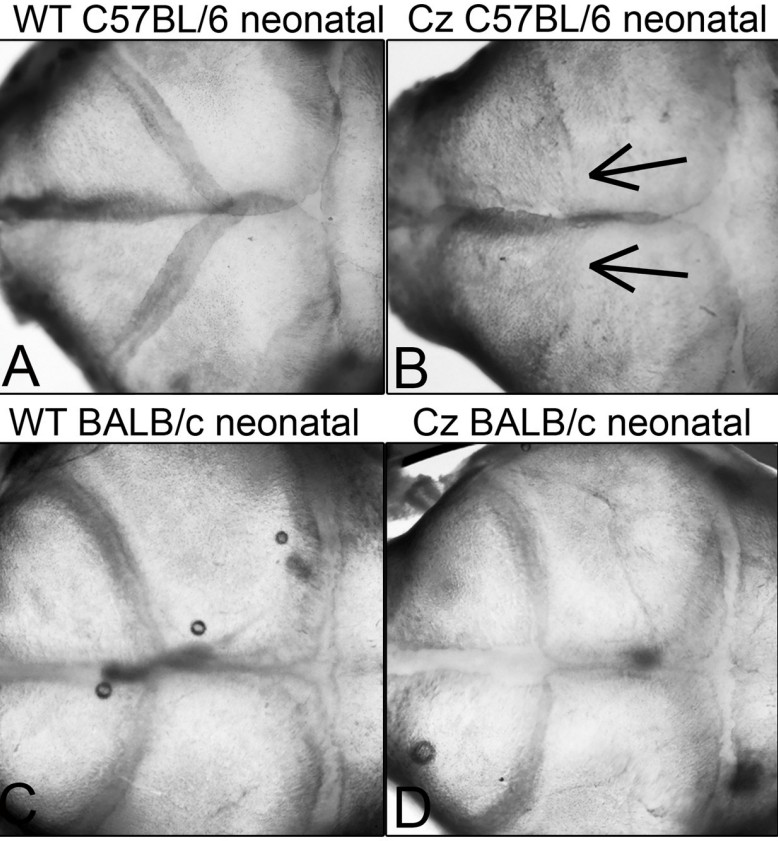

**Fig 1. Neonatal coronal suture fusion in C57BL/6 but not BALB/c FGFR2$^{C342Y/+}$ mice.** Representative images of dissected C57BL/6 FGFR2$^{+/+}$ (A), C57BL/6 FGFR2$^{C342Y/+}$ (B), BALB/c FGFR2$^{+/+}$ (C) and BALB/c FGFR2$^{C342Y/+}$ (D) post-natal day 3 mouse skulls are shown. Parietal with frontal bone overlap is evident in all shown mice. Coronal suture fusion is only present in C57BL/6 FGFR2$^{C342Y/+}$ mice. Arrows point to loss of the coronal suture in C57BL/6 FGFR2$^{C342Y/+}$ (Cz) mice.

pair band for FGFR2 and a 300 base pair band for mutant FGFR2$^{C342Y}$. Mice were fed ad libitum and housed under standard 12 hour dark/light cycles. Litters were randomly assigned to treatment/no treatment groups. Treated mice were injected with 1.0 x 10$^7$ transforming units lentivirus or an equivalent volume of phosphate buffered saline via the jugular vein after birth. BALB/c mice (n = 12 FGFR2$^{+/+}$ vehicle treated mice, n = 14 FGFR2$^{C342Y/+}$ vehicle treated mice, n = 16 FGFR2$^{C342Y/+}$ TNAP lentivirus treated mice) were euthanized by CO$_2$ overdose at four weeks post-natal and C57BL/6 mice (n = 14 FGFR2$^{+/+}$ vehicle treated mice, n = 14 FGFR2$^{C342Y/+}$ vehicle treated mice, n = 17 FGFR2$^{C342Y/+}$ TNAP lentivirus treated mice) were euthanized by CO$_2$ overdose at three weeks post-natal for analyses. BALB/c mice were sacrificed at a later age than C57BL/6 mice because craniosynostosis onset occurs later in BALB/C than in C57BL/6 FGFR2$^{C342Y/+}$ mice. Blood was collected by aortic puncture under surgical anesthesia. Mice were weighed, and body length was measured for each animal. All animal procedures were prospectively approved of by the University of Michigan's University Committee on Use and Care of Animals (UCUCA, protocol PRO00006815). All samples were de-identified as to genotype and treatment group for analyses, then unblinded for statistical comparison. The primary outcome assessment was craniosynostosis incidence. Secondary outcome assessments included malocclusion incidence, cranial bone micro CT measurements, craniofacial shape measurements, and cranial base bone lengths.

## Serum analyses

Mice were fasted for six hours prior to blood collection. Alkaline phosphatase activity (AP) in serum was quantified using the colorimetric reagent 4-nitrophenyl-phosphate disodium hexahydrate (Sigma-Aldrich), as compared to a standard curve using commercially available alkaline phosphatase enzyme (Sigma-Aldrich). One unit of AP was defined as the amount of enzyme needed to generate 1 µmol of p-nitrophenol per minute. Inorganic phosphate quantifications were performed using commercially available kits (Pointe Scientific), also as compared to standard curves. The levels of alanine amino transferase (ALT) and Aspartate amino transferase (AST) were determined using commercially available colorimetric assay kits according to manufacturer instructions (Sigma-Aldrich) using serum from control and treated BALB/c mice, as these mice yielded more serum per mouse. One unit of ALT was defined as the amount of enzyme that would generate 1 nmol of pyruvate per minute. One unit of AST was defined as the amount of enzyme that would generate 1 nmol of glutamate per minute.

## Micro Computed Tomography (micro CT)

Whole skulls were scanned at an 18 µm isotropic voxel resolution using the eXplore Locus SP micro CT imaging system (GE Healthcare Pre-Clinical Imaging, London, ON, Canada). Regions of interest (ROI's) for parietal and frontal bones were established as 0.5 mm in length, 0.5 mm in width, and depth equivalent to thickness of bone, as previously described [17, 25]. Density, volume and mineral content of cranial bones from mice were measured using previously established methods using Microview version 2.2 software (GE Healthcare Pre-Clinical Imaging, London, ON) and established algorithms [34, 35]. Linear measurements of cranial base bones were measured on micro CT scans using Dolphin imaging software (Dolphin Imaging & Management Solutions).

## Cranial suture and cranial base synchondrosis assessment

Fusions between frontal and parietal cranial bones (fusion of the coronal suture), fusion between parietal bones (fusion of the sagittal suture) and fusion of the intersphenoidal synchondrosis (ISS) and spheno-occipital synchondrosis (SOS) were identified on micro CT scans

of mouse skulls. Cranial sutures and synchondroses were viewed using the two-dimensional micro CT slices in an orthogonal view across the entire length of the suture or synchondrosis, as previously described [17, 25]. Synchondroses were identified as fused or not fused. To distinguish between more vs. less severe forms of craniosynostosis, fusion of cranial suture was scored in the following categories: 0) normal open suture, 1) diminished suture width with no fusion 2) diminished suture width with point fusions, and 3) obliteration of the suture.

Reliability of suture fusion assessment was verified by both intra-operator and inter-operator reliability statistics by calculating intraclass correlation coefficients. Intra-operator reliability statistics was carried out by assessing suture fusion status of the coronal and sagittal sutures as well as the ISS and SOS synchondroses on fifteen micro CT scans by one investigator two times separated by a two-month period. Inter-operator reliability was carried out by analyzing fifteen micro CT scans by a second investigator. The intraclass correlation coefficient for intraoperator reliability for fusion assessment is .970 (p≤.0001) and the intraclass correlation coefficient for interoperator reliability is .972 (p≤.0001). Thus, there is high intraoperator and interoperator reliability for fusion assessment.

## Linear measurements

Craniofacial linear skeletal measurements were taken using digital calipers on dissected skulls. Linear measurements were calculated using previously reported craniofacial skeletal landmarks [25, 36, 37], including standard measurements currently in use by the Craniofacial Mutant Mouse Resource of Jackson Laboratory (Bar Harbor, ME). Linear measurements were normalized to total skull length (measured from nasale to opisthion) to account for size differences between FGFR2[+/+] and FGFR2[C342Y/+] mice. Measurements were performed twice and an average of the two measurements was utilized for statistical comparison by genotype and treatment. Cranial base anterior-posterior bone lengths were measured on micro CT scans using *Dolphin Imaging 11.0* software (Dolphin Imaging and Management Solutions, Chatsworth, CA), as previously described [26].

## Statistics

Results from previous studies and this study demonstrate no sex difference for craniosynostosis in FGFR2[C342Y/+] mice, therefore sexes were combined for analyses [25]. Normality of data was evaluated by D'Agostino & Pearson Tests. Students t test was used to compare groups that were of normal distribution and the Mann Whitney test was utilized to compare groups that were of non-normal distribution. Because serum AP levels varied in mice injected with the lentivirus, linear regressions were performed to determine if, and to what extent serum AP levels associated with changes body weight and serum liver enzyme levels. Incidences of craniosynostosis severity category, synchondrosis fusion and malocclusion were analyzed by the Fisher's exact test.

## Results

Injection with the TNAP expression lentivirus significantly increased serum AP levels in all of the treated mice (Table 1). BALB/c FGFR2[C342Y/+] mice injected with the lentivirus increased serum AP levels by 1.2 U/mL when compared to BALB/c vehicle treated FGFR2[C342Y/+] mice (p<0.0001) when measured at four weeks old. C57BL/6 FGFR2[C342Y/+] mice injected with the TNAP expression lentivirus increased serum AP levels by 1.8 U/mL when compared to vehicle treated C57BL/6 FGFR2[C342Y/+] mice (p<0.0001) when measured at three weeks old. No significant difference in serum AP levels were seen between untreated FGFR2[C342Y/+] and FGFR2[+/+]

**Table 1. Serum alkaline phosphatase and inorganic phosphate measurements in vehicle vs. TNAP treated BALB/c and C57BL/6 mice.**

| Strain | Genotype | Treatment | Serum AP Level (units/ml) | Serum Pi Level (mg/dl) |
|--------|----------|-----------|---------------------------|------------------------|
| BALB/c | FGFR2$^{+/+}$ | vehicle | 0.03 +/- 0.01 | 10.9 +/- 0.8 |
| BALB/c | FGFR2$^{C342Y/+}$ | vehicle | 0.03 +/- 0.78 | 9.9 +/- 1.4 |
| BALB/c | FGFR2$^{C342Y/+}$ | TNAP | 1.30 +/- 0.54[#] | 9.8 +/- 1.1 |
| C57BL/6 | FGFR2$^{+/+}$ | vehicle | 0.01 +/- 0.01 | 9.5 +/- 1.0 |
| C57BL/6 | FGFR2$^{C342Y/+}$ | vehicle | 0.02 +/- 0.01 | 8.7 +/- 1.1 |
| C57BL/6 | FGFR2$^{C342Y/+}$ | TNAP | 1.93 +/- 0.77[#] | 9.0 +/- 0.6 |

[#] p value < 0.01 between treatment groups.

mice on the BALB/c or C57BL/6 backgrounds. Injection with the lentivirus did not alter serum inorganic phosphate (P$_i$) levels.

Statistical comparison of groups by Mann Whitney showed that FGFR2$^{C342Y/+}$ mice weigh less and are shorter in body length than their FGFR2$^{+/+}$ littermates, regardless of genetic background (Fig 2). Qualitative analysis of craniofacial skeletal shape suggested that FGFR2$^{C342Y/+}$ mice differ in morphology from their FGFR2$^{+/+}$ counterparts, and that delivery of mineral-targeted TNAP via lentivirus did not impact skull morphology (Fig 3).

Consistent with images shown in Fig 3, craniofacial skeletal linear measurements normalized to total skull length revealed many differences between FGFR2$^{342Y/+}$ and FGFR2$^{+/+}$ mice on both congenic backgrounds (Table 2). BALB/c FGFR2$^{C342Y/+}$ mice had increased cranial height, cranial width, inner canthal distance, parietal bone length and cranial height to width ratios, with decreased nasal bone length. C57BL/6 FGFR2$^{C342Y/+}$ mice had increased cranial height, cranial width, inner canthal distance, frontal bone length, parietal bone length and cranial height to width ratios, with decreased nose and nasal bone lengths. Treatment with the TNAP lentivirus did not alter craniofacial skeletal measurements in FGFR2$^{C342Y/+}$ mice on either genetic background.

Qualitative micro CT based analysis of coronal suture fusions revealed diminished suture width (grade 1) in the majority of BALB/c FGFR2$^{C342Y/+}$ vehicle treated mice with point fusions (grade 2) evident in approximately 30% of the mice. In the more severe C57BL/6 FGFR2$^{C342Y/+}$ vehicle treated mice, the majority of the mice had obliteration of the suture (grade 3) with point fusions (grade 2) evident in the rest of the mice (Fig 4). The sagittal suture was not fused in any of the mice, and no cranial suture fusions were evident in FGFR2$^{+/+}$ mice on either background. Treatment with recombinant TNAP had no significant impact on the incidence of grade 1, 2 or 3 of the coronal suture in the BALB/c FGFR2$^{C342Y/+}$ mice. While there was only a trend for decreased fusion in the left coronal suture fusion upon treatment (control 94% suture obliteration vs. treated 78% suture obliteration), there was a significant decrease in coronal suture fusion upon treatment in the right coronal suture (control 94% suture obliteration vs. treated 57% suture obliteration, p<0.03). There was also a significant decrease in coronal suture fusion when combining both right and left coronal sutures (control 94% suture obliteration vs. treated 67% suture obliteration, p<0.02). Consistent with rescue of coronal suture fusion in C57BL/6 but not BALB/c mice, the incidence of malocclusion was also significantly decreased by treatment from 82.4% to 34.7% in C57BL/6 but not BALB/c FGFR2$^{C342Y/+}$ mice (Fig 5, p<0.03).

Analysis of cranial base synchondrosis fusions revealed a 100% incidence of fusion of the inter-sphenoidal synchondrosis (ISS) in both strains of FGFR2$^{C342Y/+}$ mice, with no fusions evident in FGFR2$^{+/+}$ mice. Comparison of vehicle vs. treated mice revealed no significant change in the incidence of ISS fusion in FGFR2$^{C342Y/+}$ mice on either genetic background.

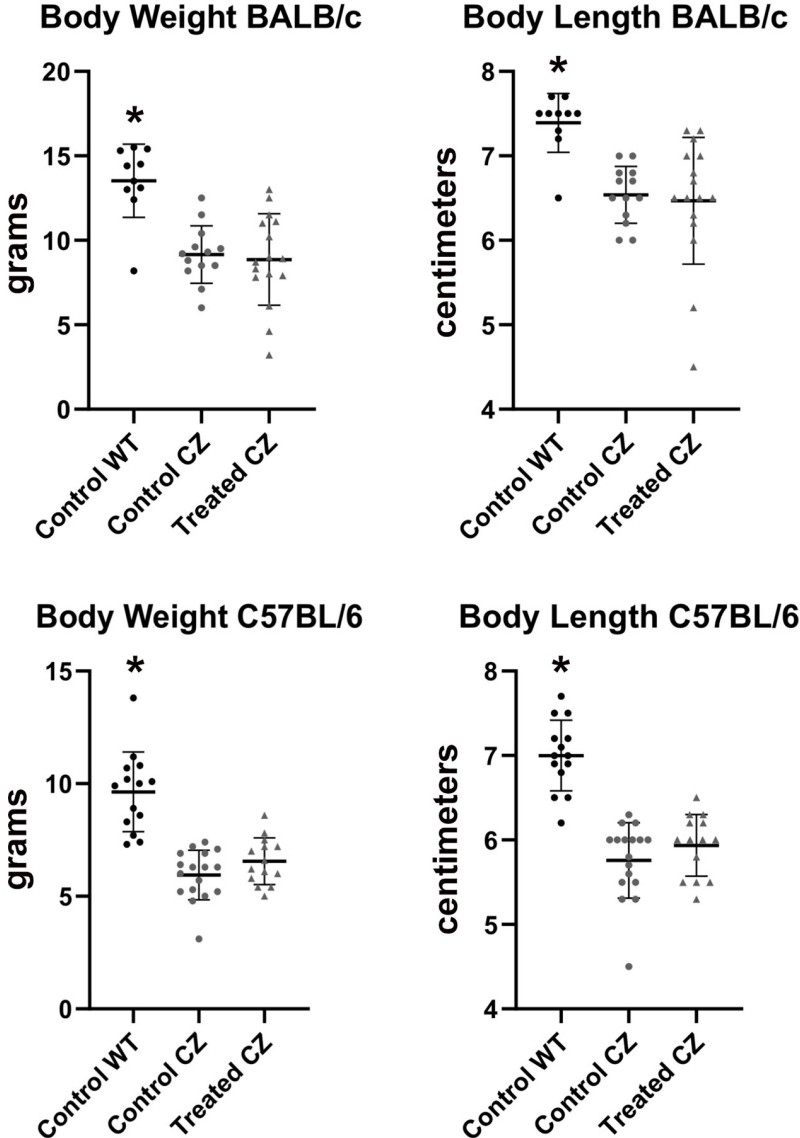

**Fig 2. Body weight and length of BALB/c and C57BL/6 mice.** Body weight and length of vehicle vs. TNAP treated mice are shown. FGFR2$^{C342Y/+}$ mice are lighter and smaller than FGFR2$^{+/+}$ mice, regardless of strain. Treatment does not alter body weight or body length. *p<0.01 between genotypes.

Spheno-occipital synchondronsis (SOS) fusion was seen in approximately half of C57BL/6 FGFR2$^{C342Y/+}$ mice but rarely in BALB/c FGFR2$^{C342Y/+}$ mice. Comparison of vehicle vs. treated mice revealed no significant change in the incidence of SOS fusion in FGFR2$^{C342Y/+}$ mice on either genetic background. Measurements of cranial base bone lengths demonstrated decreased length of the basis-sphenoid and pre-sphenoid bones in both BALB/c and C57BL/6 vehicle treated FGFR2$^{C342Y/+}$ as compared to FGFR2$^{+/+}$ mice (Table 3). Treatment increased length of the pre-sphenoid bone in BALB/c and C57BL/6 FGFR2$^{C342Y/+}$ mice, but not to the equivalent of pre-sphenoid bone length seen in FGFR2$^{+/+}$ mice.

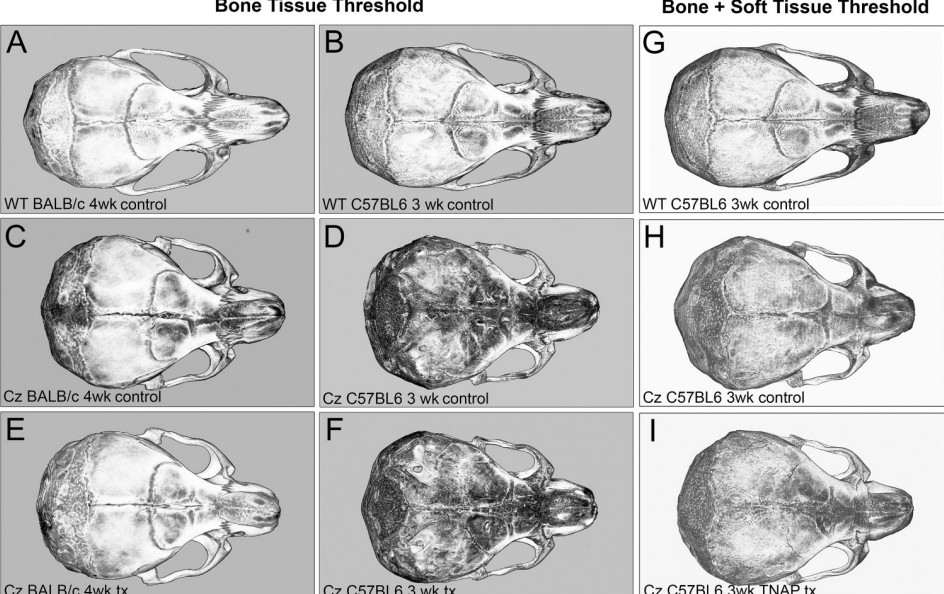

**Fig 3. Isosurface micro CT images of control and TNAP treated FGFR2$^{C342Y/+}$ mice.** Micro CT isosurface axial images of representative day 28 BALB/c and day 21 C57BL/6 mouse skulls are shown. Denser bone tissues are lighter in color. Control indicates no lentiviral delivery of TNAP. Tx indicates lentiviral delivery of TNAP. Comparison of BALB/c FGFR2$^{+/+}$ (WT) mice (A), BALB/c FGFR2$^{C342Y/+}$ (CZ) vehicle treated mice (C), and BALB/c FGFR2$^{C342Y/+}$ (CZ) TNAP treated mice (E) indicates morphologic differences between genotypes but not treatment groups. Comparison of C57BL/6 FGFR2$^{+/+}$ (WT) mice (B), C57BL/6 /c FGFR2$^{C342Y/+}$ (CZ) control mice (D), and C57BL/6 FGFR2$^{C342Y/+}$ (CZ) TNAP treated mice (F) indicates morphologic differences between genotypes but not treatment groups. (A-F) Isosurface images taken at a bone threshold. Skull images of control (D) and TNAP treated (F) C57BL/6 FGFR2$^{C342Y/+}$ mice show both cranial vault and underlying cranial base due translucent poorly mineralized cranial bones in these mice. (G-I) Isosurface images taken at a threshold that includes both bone and soft tissue are provided for C57BL/6 FGFR2$^{+/+}$ (WT) mice (G), C57BL/6 FGFR2$^{C342Y/+}$ control mice (H), and C57BL/6 FGFR2$^{C342Y/+}$ TNAP treated mice (I) are provided for improved visualization of these skulls.

Micro CT based analyses of cranial bones demonstrated significantly diminished bone mineral density, tissue mineral content, tissue mineral density and bone volume fraction in frontal bones, plus significantly diminished tissue mineral density and bone volume fraction in parietal bones of vehicle treated FGFR2$^{C342Y/+}$ mice when compared to FGFR2$^{+/+}$ littermates on both BALB/c and C57BL/6 backgrounds (Table 4). Injection with the TNAP lentivirus significantly increased frontal bone mineral density, tissue mineral content and bone volume fraction, plus parietal bone volume fraction in FGFR2$^{C342Y/+}$ mice on the BALB/c background. Injection with the TNAP lentivirus did not significantly impact any of the cranial bone parameters in FGFR2$^{C342Y/+}$ mice on the C57BL/6 background.

Biodistribution of lentivirus containing the recombinant mineral-targeted form of TNAP was previously shown to result in highest viral expression levels in the liver [30]. Because the treated mice in this study received recombinant TNAP via lentivirus, while the control group received no virus, we measured serum liver enzymes levels to determine if liver toxicity due to lentivirus could account for a change in phenotype in the treated mice, and/or the increase in serum AP levels seen in the treated mice. Alanine amino transferase (ALT) and Aspartate amino transferase (AST) are liver enzymes that, when seen at high levels in serum, are indicative of liver toxicity [38, 39]. Therefore, we tested serum ALT and AST levels in the mice. When compared by genotypes, no differences were noted between FGFR2$^{+/+}$ and control

**Table 2. Linear craniofacial skeletal measurements reveal.**

| Strain | Measurement | FGFR2$^{+/+}$ vehicle | FGFR2$^{C342Y/+}$ vehicle | FGFR2$^{C342Y/+}$ TNAP |
|---|---|---|---|---|
| BALB/c | Cranial Height | 0.36 +/- 0.01* | 0.45 +/- 0.01 | 0.45 +/- 0.01 |
| BALB/c | Cranial Width | 0.55 +/- 0.01* | 0.63 +/- 0.01 | 0.62 +/- 0.01 |
| BALB/c | Inner Canthal Distance | 0.20 +/- 0.01* | 0.25 +/- 0.01 | 0.26 +/- 0.01 |
| BALB/c | Nose Length | 0.65+/- 0.01 | 0.65 +/- 0.01 | 0.65 +/- 0.01 |
| BALB/c | Nasal Bone Length | 0.33 +/- 0.02* | 0.32 +/- 0.01 | 0.32 +/- 0.03 |
| BALB/c | Frontal Bone length | 0.33 +/- 0.02 | 0.33 +/- 0.01 | 0.34 +/- 0.03 |
| BALB/c | Parietal Bone Length | 0.20 +/- 0.01* | 0.25 +/- 0.02 | 0.26 +/- 0.01 |
| BALB/c | Ratio Height to Width | 0.66 +/- 0.02* | 0.72 +/- 0.02 | 0.72 +/- 0.02 |
| C57BL/6 | Cranial Height | 0.38 +/- 0.01* | 0.52 +/- 0.01 | 0.52 +/- 0.03 |
| C57BL/6 | Cranial Width | 0.55 +/- 0.01* | 0.64 +/- 0.01 | 0.64 +/- 0.02 |
| C57BL/6 | Inner Canthal Distance | 0.23 +/- 0.01* | 0.29 +/- 0.01 | 0.29 +/- 0.01 |
| C57BL/6 | Nose Length | 0.65 +/- 0.01* | 0.62 +/- 0.01 | 0.63 +/- .03 |
| C57BL/6 | Nasal Bone Length | 0.32 +/- 0.01* | 0.23 +/- 0.02 | 0.23 +/- 0.01 |
| C57BL/6 | Frontal Bone length | 0.35 +/- 0.01* | 0.40 +/- 0.02 | 0.41 +/- 0.03 |
| C57BL/6 | Parietal Bone Length | 0.22 +/- 0.01* | 0.26 +/- 0.02 | 0.27 +/- 0.02 |
| C57BL/6 | Ratio Height to Width | 0.38 +/- 0.01* | 0.52 +/- 0.01 | 0.52 +/- 0.03 |

No Rescue of Craniofacial Shape by TNAP Treatment in FGFR2$^{C342Y/+}$ Mice.

Measurements are reported as normalized to total skull length.

* p value < 0.01 between genotypes

No statistical differences between treatment groups were found.

FGFR2$^{C342Y/+}$ mice for either ALT or AST enzymes. When compared by treatment groups, FGFR2$^{C342Y/+}$ mice that were treated had significantly lower ALT levels than their control counterparts, and significantly higher AST levels than their control counterparts (Fig 6). We next performed correlation studies between ALT and AST serum levels with serum AP levels and body weight to better understand if lentiviral delivery of TNAP led to liver toxicity. No

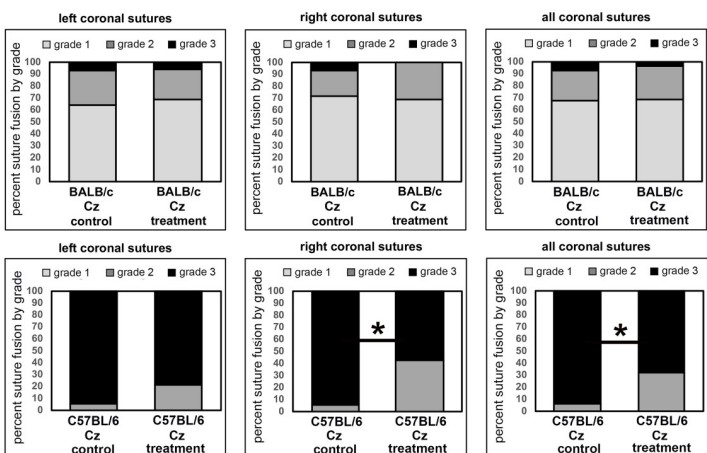

**Fig 4. Coronal suture fusion in vehicle and TNAP treated FGFR2$^{C342Y/+}$ mice.** Percentage of mice with fusion of the coronal suture are shown. Fusion was scored in the following categories: 0) normal open suture, 1) diminished suture width with no fusion, 2) diminished suture width with point fusions across the suture, and 3) obliteration of the suture. Results show diminished suture obliteration in C57BL/6 FGFR2$^{C342Y/+}$ (Cz) mice with no changes noted in BALB/c FGFR2$^{C342Y/+}$ (Cz) mice. *p<0.03 between treatment groups.

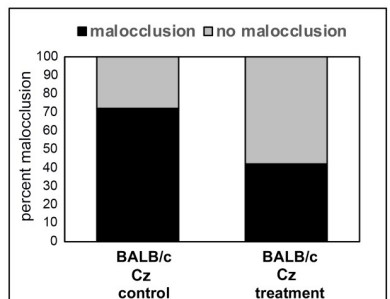
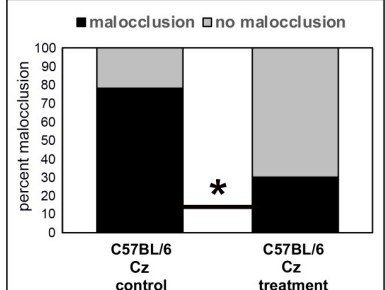

**Fig 5. Incidence of malocclusion in vehicle and TNAP treated FGFR2$^{C342Y/+}$ mice.** Percentage of FGFR2$^{C342Y/+}$ (Cz) mice with a class III malocclusion are shown. Treatment with TNAP lentivirus significantly diminished malocclusion in C57BL/6 FGFR2$^{C342Y/+}$ (Cz) but not BALB/c FGFR2$^{C342Y/+}$ (Cz) mice. *p<0.03.

significant correlations were found between serum AP and AST levels for any of the groups. A significant reverse correlation was found between serum AP and ALT levels only for the treated FGFR2$^{C342Y/+}$ group (r = -.73; 95% confidence interval of -.898 to -.361, p < .005). No significant correlations were found between serum AST or ALT levels and body weight for any of the groups. Together, these data show that the effect of lentiviral TNAP treatment decreased ALT and increased AST enzymes in serum, but had no effect on body weight of the mice and was not likely the cause of the observed increase in serum AP levels in treated mice.

## Discussion

Here we found that neonatal lentiviral delivery of recombinant TNAP increased cranial bone density, mineral content and volume fraction in the milder BALB/c FGFR2$^{C342Y/+}$ model of Crouzon syndrome but not in the more severe C57BL/6 FGFR2$^{C342Y/+}$ model of Crouzon syndrome. Increases in cranial bone density, mineral content and volume fraction by TNAP treatment in BALB/c Crouzon mice is consistent with results showing that recombinant mineral targeted TNAP treatment rescues mineralization of craniofacial and long bones in in the *Alpl$^{-/-}$* mouse model of infantile hypophosphatasia (HPP) and humans with infantile and childhood HPP [30, 31, 33, 40]. It is curious that delivery of mineral-targeted TNAP did not increase cranial bone parameters in the more severe C57BL/6 FGFR2$^{C342Y/+}$ mice. FGFR2$^{+/+}$ C57BL/6 mice have significantly less cranial bone density, mineral content and volume fraction than BALB/c FGFR2$^{+/+}$ mice, which is diminished to an even greater extent in

**Table 3. Cranial base bone measurements in vehicle vs. TNAP treated BALB/C and C57Bl/6 mice.**

| Strain | Genotype | Treatment | Basis Occipitus (mm) | Basis Sphenoid (mm) | Pre-Sphenoid (mm) |
|--------|----------|-----------|---------------------|---------------------|-------------------|
| Balb/C | FGFR2$^{+/+}$ | vehicle | 3.1 +/- 0.1 | 2.9 +/- 0.1* | 2.5 +/- 0.1* |
| Balb/C | FGFR2$^{C342Y/+}$ | vehicle | 2.9 +/- 0.3 | 2.4 +/- 0.3 | 1.8 +/- 0.1 |
| Balb/C | FGFR2$^{C342Y/+}$ | TNAP | 2.9 +/- 0.2 | 2.5 +/- 0.3 | 1.9 +/- 0.1$^{\#}$ |
| C57BL/6 | FGFR2$^{+/+}$ | vehicle | 2.8 +/- 0.2 | 2.9 +/- 0.2* | 2.2 +/- 0.1* |
| C57BL/6 | FGFR2$^{C342Y/+}$ | vehicle | 2.8 +/- 0.1 | 2.7 +/- 0.1 | 1.7 +/- 0.1 |
| C57BL/6 | FGFR2$^{C342Y/+}$ | TNAP | 3.0 +/- 0.2 | 2.6 +/-0.1 | 1.9 +/- 0.1$^{\#}$ |

* p value < 0.01 between genotypes.

# p value < 0.01 between treatment groups.

**Table 4. Cranial bone volume, density and mineral content in control and TNAP treated BALB/c and C57BL/6 mice.**

|  | Genotype | Treatment | Cranial Bone | Bone Mineral Content (mg) | Bone Mineral Density (mg/cc) | Tissue Mineral Content (mg) | Tissue Mineral Density (mg/cc) | Bone Volume Fraction |
|---|---|---|---|---|---|---|---|---|
| BALB/c | FGFR2$^{+/+}$ | vehicle | Frontal | 0.035 +/- 0.004 | 405 +/- 14* | 0.028 +/- 0.007* | 692 +/- 14* | 0.41 +/- 0.03* |
| BALB/c | FGFR2$^{C342Y/+}$ | vehicle | Frontal | 0.031 +/- 0.008 | 361 +/- 63 | 0.020 +/- 0.006# | 671 +/- 18 | 0.36 +/- 0.00 |
| BALB/c | FGFR2$^{C342Y/+}$ | TNAP | Frontal | 0.035 +/- 0.005 | 401 +/- 26# | 0.026 +/- 0.015# | 683 +/- 29 | 0.42 +/- 0.06# |
| BALB/c | FGFR2$^{+/+}$ | vehicle | Parietal | 0.034 +/- 0.004 | 405 +/- 12 | 0.023 +/- 0.005 | 693 +/- 15* | 0.43 +/- 0.03* |
| BALB/c | FGFR2$^{C342Y/+}$ | vehicle | Parietal | 0.031 +/- 0.007 | 396 +/- 39 | 0.020 +/- 0.005 | 669 +/- 24 | 0.36 +/- 0.06 |
| BALB/c | FGFR2$^{C342Y/+}$ | TNAP | Parietal | 0.034 +/- 0.006 | 403 +/- 27 | 0.025 +/- 0.012 | 691 +/- 36 | 0.42 +/- 0.07# |
| C57BL/6 | FGFR2$^{+/+}$ | vehicle | Frontal | 0.017 +/- 0.003 | 245 +/- 25* | 0.006 +/- 0.001* | 570 +/- 18* | 0.12 +/- 0.01* |
| C57BL/6 | FGFR2$^{C342Y/+}$ | vehicle | Frontal | 0.013 +/- 0.002 | 209 +/- 25 | 0.004 +/- 0.001 | 519 +/- 26 | 0.10 +/- 0.01 |
| C57BL/6 | FGFR2$^{C342Y/+}$ | TNAP | Frontal | 0.016 +/- 0.004 | 225 +/- 30 | 0.004 +/- 0.001 | 553 +/- 40 | 0.11 +/- 0.01 |
| C57BL/6 | FGFR2$^{+/+}$ | vehicle | Parietal | 0.015 +/- 0.001 | 237 +/- 19 | 0.005 +/- 0.001 | 590 +/- 13* | 0.12 +/- 0.02* |
| C57BL/6 | FGFR2$^{C342Y/+}$ | vehicle | Parietal | 0.012 +/- 0.003 | 217 +/- 18 | 0.004 +/- 0.001 | 558 +/- 36 | 0.10 +/- 0.01 |
| C57BL/6 | FGFR2$^{C342Y/+}$ | TNAP | Parietal | 0.013 +/- 0.004 | 232 +/- 42 | 0.004 +/- 0.001 | 575 +/- 32 | 1.11 +/- 0.01 |

* p value < 0.01 between genotypes.

# p value < 0.01 between treatment groups.

FGFR2$^{C342Y/+}$ C57BL/6 mice. While not quantified here, it is possible that the lack of rescue by TNAP is due to diminished bone matrix available to mineralize and/or a lack of cranial bone progenitor cells available to generate additional bone in the C57BL/6 mice, as TNAP is known to be essential for matrix mineralization [41] and for the formation of cranial bone progenitor cells [42].

Craniosynostosis severity was significantly diminished by treatment in the C57BL/6 FGFR2$^{C342Y/+}$ mice. The vast majority of these mice typically exhibit obliteration of the coronal suture within a few days after birth (Fig 1). Suture obliteration went from 94% to 67% in the C57BL/6 FGFR2$^{C342Y/+}$ mice upon treatment with exogenous TNAP by three weeks after birth. This result is consistent with the rescue of craniosynostosis seen in *Alpl*$^{-/-}$ mice treated with mineral targeted recombinant TNAP protein [33]. While far from a complete rescue, the data provided here demonstrate the potential efficacy of TNAP for diminishing severity of craniosynostosis in Crouzon syndrome. The data also support the idea that convergence exists between changes downstream of TNAP activity and FGFR2 signaling leading to coronal suture fusion. We recently showed that TNAP regulates expression of FGFR2 and Erk1,2 activity [42]. While the current study was originally based upon the hypothesis that FGF signaling regulates expression of TNAP, our more recent data suggest the alternative hypothesis that exogenous delivery of TNAP rescues Crouzon craniosynostosis because TNAP diminishes the overactive FGF and Erk1,2 signaling in FGFR2$^{C342Y/+}$ cells. We are currently working to confirm this latter hypothesis and delineate how TNAP may mediate these changes.

This study was designed to include two mouse models of Crouzon syndrome: a severe form (C57BL/6 FGFR2$^{C342Y/+}$ mice) that exhibited neonatal onset of coronal suture obliteration and a more moderate form (BALB/c FGFR2$^{C342Y/+}$ mice) that exhibited small point fusions across the coronal suture evident by four weeks after birth. We anticipated that neonatal delivery of TNAP via lentivirus would have greater effects on the later onset BALB/c FGFR2$^{C342Y/+}$ mouse model, as the drug would be delivered prior to onset of craniosynostosis. Results showed instead that the treatment significantly diminished severity of coronal suture fusion in the neonatal onset C5BL/6 FGFR2$^{C342Y/+}$ mouse with no effect on coronal suture fusion in the postnatal onset BALB/c FGFR2$^{C342Y/+}$ mouse. This data suggest that exogenous TNAP may be able to

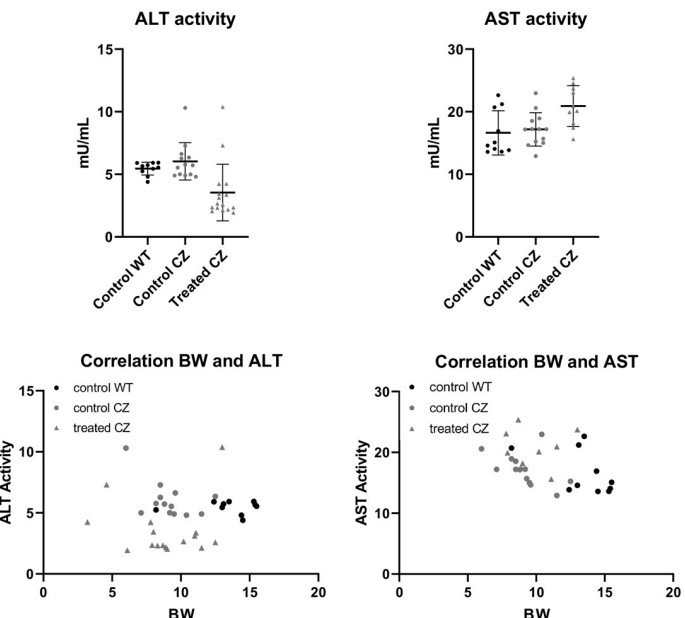

**Fig 6. Liver toxicity enzyme tests in vehicle and TNAP treated FGFR2$^{C342Y/+}$ mice.** Alanine amino transferase (ALT) and aspartate amino transferase (AST) liver enzymes were measured in serum of vehicle and treated C57BL/6 mice. When compared by groups, no differences were seen between FGFR2$^{+/+}$ and FGFR2$^{C342Y/+}$ vehicle treated mice. Treatment decreased serum ALT (A) and increased serum AST (B) in FGFR2$^{C342Y/+}$ mice. Linear regression analyses showed a significant reverse correlation (r = -.73; 95% CI of -.898 to -.361, p < .005) between AP and ALT enzyme levels in FGFR2$^{C342Y/+}$ mice (C). No correlation was found between AP and AST enzyme levels in FGFR2$^{C342Y/+}$ mice (D). No correlations were found between ALT (E) or AST (F) enzyme levels and body weight.

prevent cranial suture obliteration but not point fusions across the suture. The data also indicate that cellular mechanisms leading to craniosynostosis in FGFR2$^{C342Y/+}$ mice are both independent and dependent upon TNAP.

While lentiviral TNAP treatment did not rescue cranial base synchondrosis fusion in the FGFR2$^{C342Y/+}$ mice, length of the pre-sphenoid bone was mildly increased in the treated mice. This result is consistent with our previous study using archival lots of the lentivirus [27] which suggested changes in inferior skull morphology and may indicate that TNAP can promote cranial base growth in Crouzon syndrome. Delivery of recombinant TNAP also decreased the incidence of class III malocclusion in the C57BL/6 but not BALB/c FGFR2$^{C342Y/+}$ mice. The malocclusion rescue is likely due to changes in coronal suture fusion as opposed to changes in the cranial base, as treatment with TNAP had minimal and similar changes on cranial base synchondroses and cranial base bone lengths in both C57BL/6 and BALB/c FGFR2$^{C342Y/+}$ mice, with an effect on coronal suture fusion only in C57BL/6 FGFR2$^{C342Y/+}$ mice.

A major limitation of this study is the method and timing of TNAP delivery. Lentiviral delivery of TNAP was not ideal, as the control group did not receive lentivirus which can lead to liver toxicity and increase serum AP levels. While we found that delivery of the TNAP lentivirus decreased serum ALT and increased serum AST levels in the treated mice, there was no impact on body weight of the mice. We interpret these findings to indicate that the treatment had minimal significant impact on overall health of the mice. In addition, serum AP levels did not correlate with serum AST levels and had a reverse correlation with serum ALT levels in treated FGFR2$^{C342Y/+}$ mice. We interpret this latter data to indicate that while changes in the liver may have occurred, liver toxicity was not the cause for increased serum AP levels in the

treated mice. Yet, we cannot be certain that the lentiviral vector itself had no impact on the craniofacial skeletal phenotype of the treated FGFR2$^{C342Y/+}$ mice. A more ideal study design would incorporate empty lentiviral vector delivery to the control mice, or delivery of a recombinant TNAP protein to the mice. It would also be worthwhile to attempt prenatal delivery of TNAP, to determine if earlier treatment would have a greater impact on rescue of coronal suture fusion in C57BL/6 FGFR2$^{C342Y/+}$ mice.

It is also worth noting that earlier time points to assess changes in the coronal suture were not performed. The decreased incidence of coronal suture obliteration in C57BL/6 Crouzon mice at 3 weeks postnatal suggests that onset occurred later in these mice. Earlier time points assessing the coronal suture after treatment could have confirmed that this was indeed the case. Histologic assessments of the suture at incremental time points during treatment would enable us to determine if the treatment delayed vs. reversed suture fusion.

Earlier onset of craniosynostosis increases morbidity, as brain growth is limited earlier and for a longer duration. In addition, the extent of cranial bone fusion can influence the type of surgical intervention needed for correction. Point fusions across the suture may only require an endoscopic suturectomy procedure, while suture obliteration requires more invasive full cranial vault remodeling procedures that carry increased risk of blood loss and long operation duration [43]. If TNAP postpones or reverses fusion of the coronal suture and/or allows for less invasive surgical procedures in Crouzon syndrome, treatment with TNAP could potentially diminish morbidity. Because the C57BL/6 FGFR2$^{C342Y/+}$ Crouzon mice show neonatal onset of coronal suture fusion such that earlier intervention might be more efficacious, in future studies it might be appropriate to consider fetal delivery of TNAP. Most likely this would involve TNAP delivery using an adeno-associated virus (AAV), as opposed to the lentiviral virus utilized here, because AAV based gene delivery is considered safer due to the fact that there exists no known AAV based human diseases and because AAV rarely inserts into the genome, thereby diminishing risk for insertional mutagenesis [44]. Yet, fetal gene therapy comes with inherent risks including potential viral incorporation into germ cells leading to offspring transmission, as well as unintended disruption of developmental processes due to inappropriate timing and/or location of viral protein expression [45]. Our results overall indicate that use of an AAV based method for immediate post-birth TNAP delivery would likely diminish severity of craniosynostosis in the FGFR2$^{C342Y/+}$ mice, suggesting potential utility in human neonates with severe Crouzon syndrome.

## Acknowledgments

We thank Prof. José Luis Millán (La Jolla, CA) and Prof. Takashi Shimada (Tokyo, Japan) for providing us with the lentiviral vector expressing mineral targeted TNAP, as reported in Yamamoto et al., 2011 (30).

## Author Contributions

**Conceptualization:** Sara Dean Schutte, Nan E. Hatch.

**Data curation:** Hwa Kyung Nam, Nan E. Hatch.

**Formal analysis:** Hwa Kyung Nam, Iva Vesela, Sara Dean Schutte, Nan E. Hatch.

**Funding acquisition:** Nan E. Hatch.

**Investigation:** Hwa Kyung Nam, Sara Dean Schutte, Nan E. Hatch.

**Methodology:** Hwa Kyung Nam, Iva Vesela, Sara Dean Schutte, Nan E. Hatch.

**Project administration:** Nan E. Hatch.

**Supervision:** Hwa Kyung Nam, Nan E. Hatch.

**Validation:** Hwa Kyung Nam, Iva Vesela, Sara Dean Schutte.

**Writing – original draft:** Sara Dean Schutte, Nan E. Hatch.

**Writing – review & editing:** Hwa Kyung Nam, Iva Vesela, Nan E. Hatch.

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
