## [Decision Letter · Decision Letter 0]

6 Mar 2020

PONE-D-20-01614

Viral delivery of tissue nonspecific alkaline phosphatase diminishes craniosynostosis in one of two FGFR2C342Y/+ mouse models of Crouzon syndrome.

PLOS ONE

Dear Dr Hatch,

Thank you for submitting your manuscript to PLOS ONE. After careful consideration, we feel that it has merit but does not fully meet PLOS ONE’s publication criteria as it currently stands. Therefore, we invite you to submit a revised version of the manuscript that addresses the points raised during the review process.

There are some minor issues to address. Also please consider the addition of the histological data requested by the reviewer, and an explanation as to my caliper measures were used instead of quantitating the micro-ct.

We would appreciate receiving your revised manuscript by Apr 20 2020 11:59PM. To enhance the reproducibility of your results, we recommend that if applicable you deposit your laboratory protocols in protocols.io, where a protocol can be assigned its own identifier (DOI) such that it can be cited independently in the future. For instructions see: http://journals.plos.org/plosone/s/submission-guidelines#loc-laboratory-protocols

We look forward to receiving your revised manuscript.

Kind regards,

JJ Cray Jr., Ph.D.

Academic Editor

PLOS ONE

Journal Requirements:

1. Please ensure that your manuscript meets PLOS ONE's style requirements, including those for file naming. The PLOS ONE style templates can be found at http://www.plosone.org/attachments/PLOSOne_formatting_sample_main_body.pdf and http://www.plosone.org/attachments/PLOSOne_formatting_sample_title_authors_affiliations.pd

3. Your ethics statement must appear in the Methods section of your manuscript. If your ethics statement is written in any section besides the Methods, please move it to the Methods section and delete it from any other section. Please also ensure that your ethics statement is included in your manuscript, as the ethics section of your online submission will not be published alongside your manuscript.

Reviewers' comments:

Reviewer's Responses to Questions

**Comments to the Author**

1. Is the manuscript technically sound, and do the data support the conclusions?

Reviewer #1: Yes

Reviewer #2: Yes

2. Has the statistical analysis been performed appropriately and rigorously? 

Reviewer #1: Yes

Reviewer #2: Yes

3. Have the authors made all data underlying the findings in their manuscript fully available?

Reviewer #1: Yes

Reviewer #2: Yes

4. Is the manuscript presented in an intelligible fashion and written in standard English?

Reviewer #1: Yes

Reviewer #2: Yes

5. Review Comments to the Author

Reviewer #1: The manuscript “Tissue nonspecific alkaline phosphatase improves bone quality but does not alleviate craniosynostosis in the FGFR2C342Y/+ mouse model of Crouzon syndrome” has been revised well. In particular, the changes made to the introduction and discussion have improved the manuscript significantly. The study that aims to determine if delivery of TNAP prevents or diminishes FGFR2C342Y driven craniosynostosis. This is an interesting and worthwhile investigation seeking to identify a pharmacological treatment for craniosynostosis. Though the authors have addressed the major concerns from the previous review, a number of minor issues still need to be addressed.

Minor Revisions:

1. Abstract Line 33: The use of the word improved is vague. As you have not defined the quality/ described normal/ unaffected bone, improved is confusing. Consider changing this language to be directional, as in “increased bone volume as compared to control.”

2. Introduction line 65: “… we found statistical differences” – What does this mean? Perhaps significant would be a better word in place of statistical?

3. Introduction lines 74-76: Consider rephrasing. The way that this is written indicates one model of Crouzon syndrome, and though only one genetic target was tested here there are in fact two different models of the human disease being studied. Please rephrase the second half of this sentence “ a model of Crouzon craniosynostosis syndrome”

4. Methods section Micro Computed Tomography, Line 149: Please be consistent with referring to Micro Computed Tomography. Either always write it out or abbreviate it at the first instance. It is in different forms (written out, micro CT, micro-CT) throughout the methods, figure legends, and results.

5. Methods section Statistics, Line 186, 187: There are extra words in this sentence. Please revise carefully.

6. Results Table 1, Lines 204-205: Please be consistent with capitalization in the table title. If measurements is capitalized, then vehicle, treatment, and mice should also be capitalized. Also, be consistent with table format. Have the heading bolded all the time or none of the time.

7. Results Table 2, Lines 231-232: Please be consistent with capitalization in the table title. Also, be consistent with table format. Have the heading bolded all the time or none of the time.

8. Results, Line 249: This seems to be the first instance of the term “wild type” please use the gene instead as I believe you mean unaffected and not a true wild-type.

9. Results Table 3, Lines 272-273: Please be consistent with capitalization in the table title. Also, be consistent with table format. Have the heading bolded all the time or none of the time.

10. Results, Line 296: The term “wild type” is used again and I again believe you mean unaffected, which would make using the gene notation more appropriate.

11. Results Table 4, Lines 302-303: Please be consistent with capitalization in the table title. Also, be consistent with table format. Have the heading bolded all the time or none of the time.

12. Results Line 324-327: Please revise this sentence to the following as data is always plural, and the sentence in currently in multiple tenses: “Together, these fata show that the effect of lentiviral TNAP treatment decreased ALT and increased AST enzymes in serum, but had not effect on body weight of the mice and was not likely the cause of the observed increase in serum AP levels in treated mice.”

13. Discussion Line 330: The use of the word improved here is also vague. Consider changing this language as suggested above.

14. Discussion Line 351: delete “does”

15. Discussion Line 352: As data is always a plural (datum is singular) supports should be changes to support. Please check for agreement of this throughout your discussion, Lines 356, 368,369 need to be revised.

16: Discussion line 369: is there a functional difference between point fusions across the suture and complete obliteration? Perhaps this could factor into the discussion to some degree as a means of better situating this study within a clinical context in the treatment of craniosynostosis.

17. Figure Legends Figure 1, Line 571: As you have indicated that the fusion of the coronal suture occurs early (pre-natally) in the C57BL/6 model please indicate that the images here are from post-natal day 3 mice. Also, within Figure 1 the notation should be consistent. Either always CZ or always Crouzon, and always C57BL6 or C57Bl6. Please check that you are consistent with these notations throughout the manuscript, tables, legends, and figures.

18. Figure Legends Figure 4, Line 592: Please be consistent throughout the manuscript, particularly within figures and legends. Either use Crouzon or CZ, not both.

19. Figure 3: what does the notation no tx vs. tx stand for? This is not clear in the legend.

20. Figure 4. Please revise this figure to have on legend that includes all of the colors and display if possible all colors on each graph. Also, please use proper and consistent capitalization on titles throughout this figure and all other figures.

21. Figure 5: Why are there not titles on these graphs?

22. Figure 6: Please indicate on the graphs and or in the legend which strain was used for these experiments.

Reviewer #2: Overall this is an interesting study, but it feels unfinished, even though the work presented here is a continuation -with significant overlap- from previous work. The study can easily be improved by the addition of histology data of the coronal suture for example. If the treatment with TNAP does indeed delay or reverse the process of synostosis this could have been shown by a longitudinal study of the suture, analysing the morphology of the suture and the histology of the sutural mesenchyme.

Material and methods – Linear measurements: I don’t understand why the authors decided to perform skull measurements using a caliper when they have microCT scans available. Quantitative data using microCT scans is far superior. It would be helpful if the authors could explain why they have decided to not do this.

Material and methods – Statistics: While the authors mention a previous study that found no differences between male and female Fgfr2-C342Y mice regarding the craniosynostosis phenotype, it is currently considered bad practise to use a mixed sex cohort, especially when testing a pharmacological intervention. I recommend the authors change this in the future.

Discussion: It would be interesting if the authors could comment on the feasibility of administering TNAP in utero.

The term perinatal is not used correctly. It refers to the period shortly before to shortly after birth. In the Abstract (line 35) perinatal is used where it should be postnatal. Also, in line 67, the addition of the term perinatal to prenatal and postnatal is superfluous.

In Figure 3, it would be helpful if the authors commented on if the images are representative for their cohorts. The quality of the images in D and F (arguably the most interesting) is poor due to a likely issue with contrast. As a result the coronal sutures -or what’s left of them- are impossible to see.

In Table 3, the # referring to the non-significant differences is missing.

6. PLOS authors have the option to publish the peer review history of their article (what does this mean?). If published, this will include your full peer review and any attached files.

Reviewer #1: No

Reviewer #2: No

---

## [Author Response · Author response to Decision Letter 0]

20 Apr 2020

Response to Journal:

References are now in PLOS ONE style.

Online data repository information is now provided with the submission.

3. Your ethics statement must appear in the Methods section of your manuscript. If your ethics statement is written in any section besides the Methods, please move it to the Methods section and delete it from any other section. Please also ensure that your ethics statement is included in your manuscript, as the ethics section of your online submission will not be published alongside your manuscript.

Animal use ethics statement including prior approval by university committee, animal protocol number and type of euthanasia performed is now included in the methods section.

Response to Reviewers:

We would like to thank the reviewers again for their thorough manuscript review and thoughtful critiques. We appreciate that the reviewers agreed that we: 1) provide a technically sound manuscript in which the data support the conclusions, 2) performed an appropriate and rigorous statistical analysis, 3) made all data underlying the findings in the manuscript fully available and 4) presented the manuscript in an intelligible fashion written in standard English.

We agree with and have made relevant changes in response to both Reviewer #1 and Reviewer #2 critiques and suggestions. The only exception to this is our inability to provide histologic data at this time (due to COVID-19 based university research activity ramp down). We were fully prepared to provide histology at final time points until this ramp down. In an attempt to provide some additional data given the lack of histology, in this revision we have an additional figure showing images of dissected skulls of the mice (Fig. 4). We hope that the editor and reviewers agree that this manuscript submission is worthy of publication. 

Reviewer #1: The manuscript “Tissue nonspecific alkaline phosphatase improves bone quality but does not alleviate craniosynostosis in the FGFR2C342Y/+ mouse model of Crouzon syndrome” has been revised well. In particular, the changes made to the introduction and discussion have improved the manuscript significantly. The study that aims to determine if delivery of TNAP prevents or diminishes FGFR2C342Y driven craniosynostosis. This is an interesting and worthwhile investigation seeking to identify a pharmacological treatment for craniosynostosis. Though the authors have addressed the major concerns from the previous review, a number of minor issues still need to be addressed.

Minor Revisions:

1. Abstract Line 33: The use of the word improved is vague. As you have not defined the quality/ described normal/ unaffected bone, improved is confusing. Consider changing this language to be directional, as in “increased bone volume as compared to control.”

We agree that “improved” is vague. The word has been replaced by “increased” and “as compared to controls” has been added in the abstract.

2. Introduction line 65: “… we found statistical differences” – What does this mean? Perhaps significant would be a better word in place of statistical?

The word “statistical” has been replaced with “significant” in this sentence.

3. Introduction lines 74-76: Consider rephrasing. The way that this is written indicates one model of Crouzon syndrome, and though only one genetic target was tested here there are in fact two different models of the human disease being studied. Please rephrase the second half of this sentence “ a model of Crouzon craniosynostosis syndrome”

This phrase has been replaced with, “in BALB/c FGFR2C342Y/+ mice, a model of less severe Crouzon craniosynostosis syndrome and/or in C67BL/6 FGFR2C342Y/+ mice, a model of more severe Crouzon craniosynostosis syndrome” in the introduction section to the manuscript (lines 75-77).

4. Methods section Micro Computed Tomography, Line 149: Please be consistent with referring to Micro Computed Tomography. Either always write it out or abbreviate it at the first instance. It is in different forms (written out, micro CT, micro-CT) throughout the methods, figure legends, and results.

To eliminate inconsistencies with regard to terminology for micro computed tomography, the methods section is now titled: Micro Computed Tomography (micro CT) and all references within the text of the manuscript have been changed to “micro CT).

5. Methods section Statistics, Line 186, 187: There are extra words in this sentence. Please revise carefully.

We apologize for this confusing sentence and thank the reviewer for pointing this out. The sentence has been revised with elimination of extra words, which should clarify interpretation of text (lines 187-189).

6. Results Table 1, Lines 204-205: Please be consistent with capitalization in the table title. If measurements is capitalized, then vehicle, treatment, and mice should also be capitalized. Also, be consistent with table format. Have the heading bolded all the time or none of the time.

Thank you for catching these errors. All tables now have bolded headings. Capitalization of sentences should also now be consistent.

7. Results Table 2, Lines 231-232: Please be consistent with capitalization in the table title. Also, be consistent with table format. Have the heading bolded all the time or none of the time.

All tables now have bolded headings. Capitalization of sentences should also now be consistent.

8. Results, Line 249: This seems to be the first instance of the term “wild type” please use the gene instead as I believe you mean unaffected and not a true wild-type.

For improved clarity, “wild type” is now referred to as FGFR2+/+ throughout the manuscript. 

9. Results Table 3, Lines 272-273: Please be consistent with capitalization in the table title. Also, be consistent with table format. Have the heading bolded all the time or none of the time.

All tables now have bolded headings. Capitalization of sentences should also now be consistent.

10. Results, Line 296: The term “wild type” is used again and I again believe you mean unaffected, which would make using the gene notation more appropriate.

“wild type” is now referred to as FGFR2+/+ throughout the manuscript.

11. Results Table 4, Lines 302-303: Please be consistent with capitalization in the table title. Also, be consistent with table format. Have the heading bolded all the time or none of the time.

All tables now have bolded headings. Capitalization of sentences should also now be consistent.

12. Results Line 324-327: Please revise this sentence to the following as data is always plural, and the sentence in currently in multiple tenses: “Together, these data show that the effect of lentiviral TNAP treatment decreased ALT and increased AST enzymes in serum, but had not effect on body weight of the mice and was not likely the cause of the observed increase in serum AP levels in treated mice.”

The final sentence of results sections on ALT/AST levels (lines 346-349) has been changed to the reviewer suggested sentence (lines 395-398).

13. Discussion Line 330: The use of the word improved here is also vague. Consider changing this language as suggested above.

Thank for this reminder to avoid ambiguous terminology. The word “improved” has been changed to “increased” here. Use of the word “improved” has also been eliminated from the manuscript text (line 352). 

14. Discussion Line 351: delete “does”

This word has been deleted (line 452). 

15. Discussion Line 352: As data is always a plural (datum is singular) supports should be changes to support. Please check for agreement of this throughout your discussion, Lines 356, 368,369 need to be revised.

The word “supports” has been changed to “support” here. Grammar has also been corrected to consistently refer to data as plural in the rest of the discussion text. 

16: Discussion line 369: is there a functional difference between point fusions across the suture and complete obliteration? Perhaps this could factor into the discussion to some degree as a means of better situating this study within a clinical context in the treatment of craniosynostosis.

This is an excellent point. Discussion of surgical risk for correction of point fusions vs. suture obliteration are now included at the end of the discussion section, including an additional reference (lines 516-522). 

17. Figure Legends Figure 1, Line 571: As you have indicated that the fusion of the coronal suture occurs early (pre-natally) in the C57BL/6 model please indicate that the images here are from post-natal day 3 mice. Also, within Figure 1 the notation should be consistent. Either always CZ or always Crouzon, and always C57BL6 or C57Bl6. Please check that you are consistent with these notations throughout the manuscript, tables, legends, and figures.

The phrase “day three” has been replaced with “post-natal day 3” mice in the legend for Figure 1.

C57BL/6 is now used consistently throughout text, tables and figure legends.

18. Figure Legends Figure 4, Line 592: Please be consistent throughout the manuscript, particularly within figures and legends. Either use Crouzon or CZ, not both.

Figure legends for figures 4 and 5 now refer to Cz as opposed to Crouzon, to be consistent with other figure legends. The figures have also been changed so as to refer to Cz, not Crouzon.

19. Figure 3: what does the notation no tx vs. tx stand for? This is not clear in the legend.

The legend for figure 3 is now labeled “control” as opposed to no tx. The legend also now includes reference to the fact that “tx” refers to lentiviral TNAP delivery and control refers to no lentiviral delivery of TNAP.

20. Figure 4. Please revise this figure to have on legend that includes all of the colors and display if possible all colors on each graph. Also, please use proper and consistent capitalization on titles throughout this figure and all other figures.

Figure 5 (previously figure 4) is revised to include all colors for all grades of suture fusion. Revision of figure title also now has consistent title capitalization.

21. Figure 5: Why are there not titles on these graphs?

Figure 6 (previously figure 5) now has titles above graphs and on the vertical axis.

22. Figure 6: Please indicate on the graphs and or in the legend which strain was used for these experiments.

The strain is now indicated in the legend for figure 7 (previously figure 6).

Reviewer #2: Overall this is an interesting study, but it feels unfinished, even though the work presented here is a continuation -with significant overlap- from previous work. The study can easily be improved by the addition of histology data of the coronal suture for example. If the treatment with TNAP does indeed delay or reverse the process of synostosis this could have been shown by a longitudinal study of the suture, analysing the morphology of the suture and the histology of the sutural mesenchyme.

We agree that it would have been ideal to do analyses at earlier time points of the treatment to determine if treatment delayed or reversed fusion. Had this reviewer suggestion been provided in the critique of the 1st manuscript submission, we would have euthanized mice at internal time points to perform histology while we were generating additional control and treated mice to obtain power for the 2nd submission. At this point, we hope that expanded discussion of this limitation is adequate for publication (lines 427-428). 

Additionally, we had planned to provide histologic sections of animals from groups at the final time points for this submission. Unfortunately, in this time of COVID-19, we have been unable to pursue these studies in a timely manner to meet the resubmission deadline. We are fully prepared to provide histology of bone and suture, but will not be able to do so until our university again allows for research activity. If histology is deemed necessary, we will proceed but cannot now say when that can be completed. 

In an attempt to address this critique under current circumstances in which we cannot pursue histology, this revision includes an additional figure (Figure 4) showing the coronal suture in dissected C57BL/6 mouse skulls. We recognize that this is not the same as histology but, given the poor bone mineralization and therefore difficulty visualizing the micro CT images in Figure 3, we thought inclusion of Figure 4 would help with visualization of the rescued coronal suture in a TNAP treated mouse. 

Material and methods – Linear measurements: I don’t understand why the authors decided to perform skull measurements using a caliper when they have microCT scans available. Quantitative data using microCT scans is far superior. It would be helpful if the authors could explain why they have decided to not do this.

We did not include micro CT based linear methods because we have done both micro CT based linear measurements plus digital caliper based measurements for numerous mouse models and find that measurements from the two methods yield similar results. So now we have taken to commonly providing only digital caliper based measurements because this method produces reproducible, high quality, quantifiable data that can provide phenotype information efficiently and prior to micro CT.

Material and methods – Statistics: While the authors mention a previous study that found no differences between male and female Fgfr2-C342Y mice regarding the craniosynostosis phenotype, it is currently considered bad practice to use a mixed sex cohort, especially when testing a pharmacological intervention. I recommend the authors change this in the future.

Thank you for this suggestion. We will proceed in the future by not using a mixed sex cohort.

Discussion: It would be interesting if the authors could comment on the feasibility of administering TNAP in utero.

Discussion of in utero TNAP delivery is now included (lines 432-444).

The term perinatal is not used correctly. It refers to the period shortly before to shortly after birth. In the 

Abstract (line 35) perinatal is used where it should be postnatal. Also, in line 67, the addition of the term perinatal to prenatal and postnatal is superfluous.

Perinatal has been replaced with the term postnatal in the abstract (line 35) and perinatal has been removed from the introduction section describing craniosynostosis onset (line 67).

In Figure 3, it would be helpful if the authors commented on if the images are representative for their cohorts. 

Figure 3 legend now includes reference to the fact that mice in these images are representative of their cohorts.

In addition, introduction to the craniofacial linear measurement data now includes the phrase, “Consistent with images shown in Fig. 3” (line 243).

The quality of the images in D and F (arguably the most interesting) is poor due to a likely issue with contrast. As a result the coronal sutures -or what’s left of them- are impossible to see.

It is true that isosurface micro CT images of the C57BL/6 FGFR2C342Y/+ mice in figure 3 (D,F) are difficult to interpret for coronal suture fusion. This is because the cranial bones of these mice are so poorly mineralized that you can see through them to the cranial base, such that the cranial base and cranial vault appear superimposed when the image is taken at a bone limited threshold. To address this concern, we now also include micro CT isosurface images of the C57BL/6 FGFR2C342Y/+ mice at a lower threshold that includes both bone and soft tissue. Please note that we did not include comments on coronal suture fusion assessment with these images, because it is not appropriate to do so. These images provide overall skull morphology.

In Table 3, the # referring to the non-significant differences is missing.

Thank you for noting this. “# p value < 0.01 between treatment groups” is now written under Table 3.

---

## [Decision Letter · Decision Letter 1]

12 May 2020

PONE-D-20-01614R1

Viral delivery of tissue nonspecific alkaline phosphatase diminishes craniosynostosis in one of two FGFR2C342Y/+ mouse models of Crouzon syndrome.

PLOS ONE

Dear Dr Hatch,

Thank you for submitting your manuscript to PLOS ONE. After careful consideration, we feel that it has merit but does not fully meet PLOS ONE’s publication criteria as it currently stands. Therefore, we invite you to submit a revised version of the manuscript that addresses the points raised during the review process.

There are a couple of outstanding point with the figures that should be addressed prior to final approval.

We would appreciate receiving your revised manuscript by Jun 26 2020 11:59PM. To enhance the reproducibility of your results, we recommend that if applicable you deposit your laboratory protocols in protocols.io, where a protocol can be assigned its own identifier (DOI) such that it can be cited independently in the future. For instructions see: http://journals.plos.org/plosone/s/submission-guidelines#loc-laboratory-protocols

We look forward to receiving your revised manuscript.

Kind regards,

JJ Cray Jr., Ph.D.

Academic Editor

PLOS ONE

Reviewers' comments:

Reviewer's Responses to Questions

**Comments to the Author**

1. If the authors have adequately addressed your comments raised in a previous round of review and you feel that this manuscript is now acceptable for publication, you may indicate that here to bypass the “Comments to the Author” section, enter your conflict of interest statement in the “Confidential to Editor” section, and submit your "Accept" recommendation.

Reviewer #1: All comments have been addressed

Reviewer #2: (No Response)

2. Is the manuscript technically sound, and do the data support the conclusions?

Reviewer #1: Yes

Reviewer #2: Yes

3. Has the statistical analysis been performed appropriately and rigorously? 

Reviewer #1: Yes

Reviewer #2: Yes

4. Have the authors made all data underlying the findings in their manuscript fully available?

Reviewer #1: Yes

Reviewer #2: Yes

5. Is the manuscript presented in an intelligible fashion and written in standard English?

Reviewer #1: Yes

Reviewer #2: Yes

6. Review Comments to the Author

Reviewer #1: The authors of this manuscript have done a very careful revision. I would like to thank them for their careful attention to consistency of language and their revision of word choice away from vague terms. I have very few minor comments as follow:

Line 138-139: There are extra words in this sentence. I suggest deleting “as the.”

Line 168-169: Please spell out ICC

Table 2: If possible please add the ones place digit for all values (0.12 instead of .12)

Table 4: If possible please add the ones place digit for all values (0.12 instead of .12). If not possible, please be consistent and never have the ones place digit included when it is 0.

Line 640: Missing word, revise to “underlying cranial base due to translucent poorly mineralized”…

Figure 4 legend: Please add “Tx indicates delivery of TNAP” to the legend.

Reviewer #2: The authors have addresses all of the minor and the paper -especially the Discussion- has improved.

However, the two main concerns are still left somewhat unresolved.

• Regarding caliper versus microCT measurements of skull dimensions, the authors claim that they “…find that measurements from the two methods yield similar results.” If this is the case I would have expected the authors to either show this as part of this paper or refer to previously published work. Mainly because this is counter-intuitive and because this will be very useful information for colleagues performing similar analyses in other laboratories. However, I will take the authors on their word that this is indeed the case.

• The paper’s main finding is that “Viral delivery of tissue nonspecific alkaline phosphatase diminishes craniosynostosis…” (from the title). This is supported by data in Figure 3 and 5, which highlights the effects of TNAP treatment on C57BL/6 Crouzon mice. I hope the authors agree that scoring the patency of the suture by looking at the external surface only tells part of the story, hence my request of an histological analysis of the underlying suture pathology. The new data in Figure 4 does* not add anything to support the effect of TNAP on the coronal suture and I suggest the authors remove this figure. The new images in Figure 3 are a useful addition, but I would like to see the same image (using the soft tissue threshold) for the WT to make it possible to compare the ‘rescued’ C57BL/6 suture with the WT one.

Under the circumstances I am prepared to accept the paper without the histological data (partly because of the improved Figure 3), but I am very much looking forward to a follow-up study that analyses the impact of TNAP treatment on the coronal suture bone and mesenchyme at the cellular and molecular level. This study represent some extremely interesting findings and it deserves a more thorough investigation.

Minor points

• Figure 1 Please include an image of the BALB/c strain at the same time point if available. I didn’t spot this in the previous submission, but it would help to highlight the difference between the two Crouzon mouse strains here.

• Table 2, line 255 “Measures are reported as normalized to total skull length.” Measures should be measurements.

• Table 3, line 312 “No significant differences between treatment groups were found.” I suspect this should be deleted.

*in my opinion (and the Oxford English Dictionary’s) the word data should be treated as a mass noun and thus takes a singular verb.

7. PLOS authors have the option to publish the peer review history of their article (what does this mean?). If published, this will include your full peer review and any attached files.

Reviewer #1: No

Reviewer #2: No

---

## [Author Response · Author response to Decision Letter 1]

16 May 2020

Response to Reviewer comments:

Reviewer #1: The authors of this manuscript have done a very careful revision. I would like to thank them for their careful attention to consistency of language and their revision of word choice away from vague terms. I have very few minor comments as follow:

Line 138-139: There are extra words in this sentence. I suggest deleting “as the.” 

This word was deleted.

Line 168-169: Please spell out ICC.

ICC is now spelled out as intraclass correlation coefficient.

Table 2: If possible please add the ones place digit for all values (0.12 instead of .12) 

Thank you for pointing this out. A digit was added at the ones place for all values.

Table 4: If possible please add the ones place digit for all values (0.12 instead of .12). If not possible, please be consistent and never have the ones place digit included when it is 0. 

Thank you for pointing this out. A digit was added at the ones place for all values.

Line 640: Missing word, revise to “underlying cranial base due to translucent poorly mineralized”… 

Thank you for catching this. The phrase was revised for this submission.

Figure 4 legend: Please add “Tx indicates delivery of TNAP” to the legend. 

This figure was removed per Reviewer #2 request.

Reviewer #2: The authors have addressed all of the minor and the paper -especially the Discussion- has improved. However, the two main concerns are still left somewhat unresolved.

• Regarding caliper versus microCT measurements of skull dimensions, the authors claim that they “…find that measurements from the two methods yield similar results.” If this is the case I would have expected the authors to either show this as part of this paper or refer to previously published work. Mainly because this is counter-intuitive and because this will be very useful information for colleagues performing similar analyses in other laboratories. However, I will take the authors on their word that this is indeed the case.

We agree with reviewer #2 that there are situations in which a micro CT based morphologic analysis is appropriate, particularly (for example) if there is a need for performing shape analyses that require 3D coordinate data of landmarks, or analyses of specific regions of the skull that cannot be assessed using caliper measurements on dissected skulls. For example, in this manuscript we measured cranial base bones on micro CT images because these measurements cannot be performed well on dissected skulls. 

That said, we have found that digital caliper measurements of the overall skull, if done accurately, can relatively quickly convey differences in craniofacial shape, and that data from digital caliper measurements reflects data from micro CT based measurements. We first discovered this working with the Alpl-/- mouse model of infantile hypophosphatasia. The craniofacial bones of Alpl-/- mice are present but severely under-mineralized, making visualization of some skeletal landmarks difﬁcult on micro computed tomographic images. Therefore, at that time we proceeded with digital caliper measurements using landmarks that included ﬁve standard measurements in use by the Craniofacial Mutant Mouse Resource of the Jackson Laboratory (Bar Harbor, ME). Results demonstrated that Alpl-/- mice are acrocephalic (taller) and brachycephalic (wider) relative to anterior-posterior length when compared to Alpl+/+ mice.1 We subsequently performed a more comprehensive morphologic analysis of the same set of mice using micro CT. Results again showed that Alpl-/- mice are acrocephalic (taller) and brachycephalic (wider) relative to anterior-posterior length when compared to Alpl+/+ mice, and that only those Alpl-/- mice with a severe bone hypomineralization defect develop the abnormal craniofacial shape.2 We have now performed digital caliper and micro CT based measurements on additional mouse models and find that conclusions of results from both methods are similar. We find that the most important factor in reporting accurate skull morphologic differences is appropriate normalization for mouse/skull size differences (regardless of method used to create craniofacial linear measurements). As primary author, I am extremely confident that use of the digital calipers for linear measurements is appropriate in this context, and that the skull morphology data reported in this manuscript accurately reflects what is seen in the mice. 

1. Liu J, Nam HK, Campbell C, Gasque KC, Millan JL, Hatch NE. Tissue-nonspecific alkaline phosphatase deficiency causes abnormal craniofacial bone development in the Alpl(-/-) mouse model of infantile hypophosphatasia. Bone 2014;67:81-94.

2. Durussel J, Liu J, Campbell C, Nam HK, Hatch NE. Bone mineralization-dependent craniosynostosis and craniofacial shape abnormalities in the mouse model of infantile hypophosphatasia. Dev Dyn 2016;245:175-182.

• The paper’s main finding is that “Viral delivery of tissue nonspecific alkaline phosphatase diminishes craniosynostosis…” (from the title). This is supported by data in Figure 3 and 5, which highlights the effects of TNAP treatment on C57BL/6 Crouzon mice. I hope the authors agree that scoring the patency of the suture by looking at the external surface only tells part of the story, hence my request of a histological analysis of the underlying suture pathology. 

The new data in Figure 4 does* not add anything to support the effect of TNAP on the coronal suture and I suggest the authors remove this figure. 

We agree that Figure 4 does not add additional supportive data. We provided it because the images were available during this time in which we cannot return to the laboratory to generate additional data. For this revision, Figure 4 was removed from the manuscript.

The new images in Figure 3 are a useful addition, but I would like to see the same image (using the soft tissue threshold) for the WT to make it possible to compare the ‘rescued’ C57BL/6 suture with the WT one.

In response to this request, an image of a representative WT C57BL/6 mouse skull using the soft tissue threshold is now included in Figure 3.

Under the circumstances I am prepared to accept the paper without the histological data (partly because of the improved Figure 3), but I am very much looking forward to a follow-up study that analyses the impact of TNAP treatment on the coronal suture bone and mesenchyme at the cellular and molecular level. This study represent some extremely interesting findings and it deserves a more thorough investigation.

This author wants to thank Reviewer #2 for their appreciation both of the significance of results shown here, and of how the current laboratory closures are negatively impacting our ability to generate additional data. We are very interested in understanding how TNAP and FGF signaling interact to control craniofacial development and will most certainly report more mechanistic cell and tissue data in future manuscripts. 

Minor points

• Figure 1 Please include an image of the BALB/c strain at the same time point if available. I didn’t spot this in the previous submission, but it would help to highlight the difference between the two Crouzon mouse strains here.

In response to this request, Figure 1 now includes images of the BALB/c strain at the same time point. 

• Table 2, line 255 “Measures are reported as normalized to total skull length.” Measures should be measurements.

Thank you for catching this typo. It was corrected for this resubmission.

• Table 3, line 312 “No significant differences between treatment groups were found.” I suspect this should be deleted.

This line was deleted.

*in my opinion (and the Oxford English Dictionary’s) the word data should be treated as a mass noun and thus takes a singular verb.

Controversy regarding use of data as singular or plural has been ongoing for the past century. All verbs for data were previously revised based upon reviewer #1’s critique of a previous resubmission (“Discussion Line 352: As data is always a plural (datum is singular) supports should be changes to support. Please check for agreement of this throughout your discussion, Lines 356, 368,369 need to be revised”). 

This author is happy to treat the term data as singular or plural, if the reviewers can agree on which it should be. For now, in this 3rd resubmission, data remains treated as plural, based upon earlier reviewer #1 feedback.

---

## [Editor Report · Decision Letter 2]

19 May 2020

Viral delivery of tissue nonspecific alkaline phosphatase diminishes craniosynostosis in one of two FGFR2C342Y/+ mouse models of Crouzon syndrome.

PONE-D-20-01614R2

Dear Dr. Hatch,

We are pleased to inform you that your manuscript has been judged scientifically suitable for publication and will be formally accepted for publication once it complies with all outstanding technical requirements.

With kind regards,

JJ Cray Jr., Ph.D.

Academic Editor

PLOS ONE
---

## [Editor Report · Acceptance letter]

21 May 2020

PONE-D-20-01614R2 

Viral delivery of tissue nonspecific alkaline phosphatase diminishes craniosynostosis in one of two FGFR2^C342Y/+^ mouse models of Crouzon syndrome. 

Dear Dr. Hatch:

I am pleased to inform you that your manuscript has been deemed suitable for publication in PLOS ONE. Congratulations! Your manuscript is now with our production department. 

With kind regards,

on behalf of

Dr. JJ Cray Jr. 

Academic Editor

PLOS ONE